# Prophylactic Medication during Radioembolization in Metastatic Liver Disease: Is It Really Necessary? A Retrospective Cohort Study and Systematic Review of the Literature

**DOI:** 10.3390/diagnostics13243652

**Published:** 2023-12-12

**Authors:** Manon N. G. J. A. Braat, Sander C. Ebbers, Ahmed A. Alsultan, Atal O. Neek, Rutger C. G. Bruijnen, Maarten L. J. Smits, Joep de Bruijne, Marnix G. E. H. Lam, Arthur J. A. T. Braat

**Affiliations:** 1Department of Radiology and Nuclear Medicine, University Medical Center Utrecht, 3508 GA Utrecht, The Netherlandsm.lam@umcutrecht.nl (M.G.E.H.L.); a.j.a.t.braat@umcutrecht.nl (A.J.A.T.B.); 2Department of Gastroenterology and Hepatology, University Medical Center Utrecht, 3508 GA Utrecht, The Netherlands

**Keywords:** radioembolization, hepatotoxicity, drug-induced liver disease, prophylaxis

## Abstract

Purpose: Trans-arterial radioembolization is a well-studied tumoricidal treatment for liver malignancies; however, consensus and evidence regarding periprocedural prophylactic medication (PPM) are lacking. Methods: A single-center retrospective analysis from 2014 to 2020 was performed in patients treated with ^90^Y-glass microspheres for neuroendocrine or colorectal liver metastases. Inclusion criteria were the availability of at least 3 months of clinical, biochemical, and imaging follow-up and post-treatment ^90^Y-PET/CT imaging for the determination of the whole non-tumorous liver absorbed dose (D_h_). Logistic regression models were used to investigate if variables (among which are P/UDCA and D_h_) were associated with either clinical toxicity, biochemical toxicity, or hepatotoxicity. Additionally, a structured literature search was performed in November 2022 to identify all publications related to PPM use in radioembolization treatments. Results: Fifty-one patients received P/UDCA as post-treatment medication, while 19 did not. No correlation was found between toxicity and P/UDCA use. D_h_ was associated with biochemical toxicity (*p* = 0.05). A literature review resulted in eight relevant articles, including a total of 534 patients, in which no consistent advice regarding PPM was provided. Conclusion: In this single-center, retrospective review, P/UDCA use did not reduce liver toxicity in patients with metastatic liver disease. The whole non-tumorous liver-absorbed dose was the only significant factor for hepatotoxicity. No standardized international guidelines or supporting evidence exist for PPM in radioembolization.

## 1. Introduction

Trans-arterial radioembolization is a well-studied tumor-reductive treatment for primary liver malignancies and liver metastases and has been proven to be safe and effective [1]. The purpose of periprocedural prophylactic medication (PPM) in radioembolization is to ensure comfort and minimize side effects such as post-embolization syndrome, or potential complications like radioembolization-induced liver disease (REILD). However, only a limited number of studies have investigated the actual efficacy of PPM in radioembolization, while several randomized controlled trials (SIRFLOX, FOXFIRE, FOXFIRE Global, SARAH, SIRVENIB, SORAMIC, EPOCH, and DOSISPHERE-1) did not use or mention the use of standard PPM [1,2,3,4,5,6,7].

Histopathological features of REILD are largely compatible with sinusoidal obstruction syndrome (SOS) [8,9]. Lodging of microspheres into the liver sinusoids results in radiation-induced changes to the normal liver parenchyma [8,9]. As a consequence, varying degrees of hepatotoxicity will develop, depending on the extent of non-tumorous liver parenchyma involvement and the presence of an underlying disease, ranging from clinically occult biochemical changes to symptomatic REILD [8,10]. Symptomatic REILD is a seriously debilitating, potentially lethal condition, but fortunately rarely reported (0–5% in large series) [11,12].

Differences in patient care amongst radioembolization centers exist, as there is no evidence-based international standard to adhere to [11,13]. In recent CIRSE questionnaires, PPM use was highly variable, both pre and post treatment. A minority of centers did not use PPM, while the remaining centers prescribed a variety of PPM in different combinations and doses (steroids, proton pump inhibitors, anti-emetics, analgesics, and antibiotics) [11,13]. 

Based on a previous publication by Gil-Alzugaray et al., the use of prednisolone with ursodeoxycolic acid (P/UDCA) was introduced in several centers [14]. Conclusive evidence for this approach is, however, lacking, and recent guidelines do not mention the use of PPM [15]. To this end, we performed a retrospective, single-center cohort study to analyze the efficacy of P/UDCA and evaluate any relevant variables that should be taken into consideration in relation to posttreatment toxicity. Additionally, a structured literature search of all available evidence concerning general PPM in radioembolization is provided.

## 2. Methods

### 2.1. Cohort Study

Data were collected on all consecutive patients treated with radioembolization from 2014 to October 2020. Retrospective analyses included patients with progressive liver dominant or liver-only colorectal liver metastases (CRLM) or with neuroendocrine liver metastases (NELM), treated with ^90^Y-glass microspheres (Therasphere^®^, Boston Scientific, Marlborough, MA, USA) as monotherapy (i.e., no concurrent systemic treatments). Including both CRLM and NELM would allow for the analysis of differences between hypo- and hypervascular disease. Hepatocellular carcinoma patients were not analyzed to avoid confounding by underlying liver disease. From 2016 to mid-2019, based on a previous publication from Gil-Alzugaray et al., P/UDCA was standard care in our center, consisting of ursodeoxycholic acid 600 mg daily for two months and prednisone 10 mg daily for one month, followed by prednisone 5 mg daily for the second month, all starting the day of treatment [14]. Prior to 2016 and from mid-2019 onwards, no P/UDCA was prescribed.

The main inclusion and exclusion criteria were baseline imaging and follow-up imaging at three months with either positron emission computed tomography/computed tomography (PET/CT) or multiphase contrast enhanced CT (CECT), full medical history (i.e., baseline and posttreatment clinical and laboratory adverse events), and availability of post-treatment ^90^Y-PET/CT for dosimetric analysis. 

Simplicit^90^y software (Mirada Medical Ltd., Oxford, UK; Version 2.4.0.43951 (64 bit)) was used for the dosimetric analysis. The ^90^Y PET/CT was registered as the last available diagnostic contrast-enhanced CT for liver and tumor delineation. The whole non-tumorous liver was considered as one volume, including both treated and non-treated parts, to calculate the whole non-tumorous liver absorbed dose (D_h_) [16].

Variables gathered were baseline patient characteristics (age, sex, World Health Organization performance status (WHO)), information on general PPM, treatment strategy (e.g., whole liver, lobar, or selective approach), clinical adverse events measured in Common Terminology Criteria for Adverse Events (CTCAE) grades (version 5.0), biochemical toxicity, prescribed average absorbed dose to the treated volume, D_h_ in Gray (Gy), previous systemic therapies, and previous liver-directed therapies. 

REILD was defined as a symptomatic post-radioembolization deterioration in the ability of the liver to maintain its (normal or preprocedural) synthetic, excretory, and detoxifying functions according to Braat et al.; characterized by jaundice and the development of or increase in ascites, hyperbilirubinemia, and hypoalbuminemia developing at least 2 weeks to 4 months after treatment, in the absence of tumor progression or biliary obstruction” [17]. 

The medical ethics committee of our institution waived the need for informed consent for review of the data.

### 2.2. Procedures

Patients’ health status at baseline was established during pre-treatment consultations and pre-treatment simulations. Pre-treatment simulation consisted of hepatic angiography and administration of technetium-99m-macroaggregated albumin (^99m^Tc-MAA) to exclude extrahepatic depositions of activity. Subsequently, therapeutic activity was calculated according to the so-called ‘MIRD formula’ (i.e., using an average absorbed dose to the treated volume), as prescribed by the manufacturer in the instructions for use. On the day of treatment, laboratory tests (i.e., bilirubin, alkaline phosphatase, aspartate aminotransferase, alanine aminotransferase, gamma-glutamyltransferase, lactate dehydrogenase, and albumin) were performed as baseline measurements. Within 24 h after treatment, a ^90^Y-PET/CT was acquired to assess dose distribution. One and three months after radioembolization, patients were seen at an out-patient clinic with laboratory testing, and three months after radioembolization with evaluation imaging. 

### 2.3. Statistical Analysis

Descriptive analyses were used to identify patient demographics and treatment characteristics. Three outcome measures were defined: clinical and biochemical toxicities were scored according to CTCAE version 5.0, and hepatotoxicity was scored according to Braat et al. [17]. For the clinical and biochemical toxicity outcome measures, a point total system was used to find a pattern, as most patients often only experience grade 1 events, but potentially on several fronts (e.g., fatigue, vomiting, and fever, all grade 1). Thus, a point total system (i.e., summed CTCAE cores) was assumed to give a better representation of patient data (as opposed to dichotomizing separate toxicities only). CTCAE scores were corrected for the pre-treatment presence of CTCAE grades, i.e., the highest CTCAE grade post-treatment was included if it was higher than the CTCAE grade pre-treatment. However, if the pre-treatment grade was equal to or greater than the post-treatment grade, these toxicities were deemed unrelated to radioembolization and excluded from the analysis. In order to dichotomize the outcome measures, summed clinical and summed biochemical CTCAE scores were defined as ≤4 vs. >4. For the hepatotoxicity score, according to Braat et al., results were dichotomized to <3 vs. ≥3 (i.e., without or with medical intervention) [17]. The three outcome measures were tested independently: the hepatotoxicity score is specifically designed for REILD only (encompassing clinical and biochemical toxicities and clinical follow-up), clinical toxicities are more generic (but affected by subjective physician reporting in a retrospective study), and biochemical toxicities are observer-independent. 

Individual variables were tested for collinearity by Spearman rank testing: WHO-performance status, tumor type, liver burden, D_h_, treatment approach (i.e., whole liver yes/no), previous chemotherapy, previous PRRT, and liver-directed therapies. Finally, the remaining non-correlated variables were dichotomized (in distinct categories or divided median-based) and assessed in binary uni- and multivariate logistic regression models to investigate a possible relationship with either of the three outcome measures. The findings were deemed statistically significant with a *p*-value of <0.05.

### 2.4. Literature Search

A PubMed search was performed in November 2022 with the following key terms, along with all their respective variations, synonyms, combinations, and MeSh terms: liver, radioembolization or selective internal radiation therapy (SIRT), octreotide, anti-emetics, antibiotics, analgesics, periprocedural, prophylactic, and prophylaxis. The full search terminology can be found in Appendix A.

Studies were included if patients were treated with radioembolization, including all commercially available particles (i.e., ^90^Y-glass, ^90^Y-resin, or ^166^Ho), and focused on pre-, peri-, or postprocedural medication. Records of adverse events and follow-up of at least three months were required. Studies on hepatocellular carcinoma patients alone were excluded. 

Careful notes were taken of factors such as dosimetry, the number of radioembolization procedures, concomitant chemotherapy, the number of adverse events and their respective CTCAE grades, and, finally, any relevant medical history noted in the articles. These factors were not mandatory but were used qualitatively (not standardized) to assess the quality of the included studies.

## 3. Results

### 3.1. Retrospective Cohort Study

#### 3.1.1. Patient Characteristics

Seventy patients were included, 57% with CRLM and 43% with NELM. The study population was extensively pre-treated (Table 1) and mostly treated in a salvage setting. The median prescribed average absorbed dose to the treatment volume was 120 Gy (range: 30–300 Gy) for the entire population, including 54% whole-liver treatments (60% in mCRC and 45% in NELM). The median interval from calibration to therapy was 4 days (range 2–11 days). The median D_h_ was 58 Gy (range 5–139 Gy). 

No patients were lost to follow-up within the first three months, and there were no missing clinical data. Three patients had partial laboratory testing at one-month follow-up, while one other patient missed the three-month laboratory testing.

#### 3.1.2. Prophylactic Medication 

Different combinations of PPM were given to patients (Table 2). Fifty-one patients received P/UDCA, while 19 patients did not. No specific PPM was prescribed for the CRLM patients. In the NELM population, none of the patients received a periprocedural octreotide infusion or an additional octreotide bolus. Five patients had a history of a biliary intervention (three biliodigestive anastomoses and two biliary stents), of which two patients received prophylactic antibiotics (one with a biliodigestive anastomosis and one with a biliary stent). In three diabetic patients, prednisolone was consciously discarded to avoid hyperglycemias/diabetic dysregulation during follow-up (4%), but they were included in the P/UDCA group.

#### 3.1.3. Toxicity

New clinical toxicities resulting from the treatment were grade ≥3 in 2% (Table 3). New CTCAE grade ≥3 biochemical toxicities occurred 30 times in 21 patients (30%) (Table 3). 

In total, 14 patients developed a hepatotoxicity score of ≥2. Hepatoxicity grade ≥3 (REILD) was diagnosed in two patients (3%) [17]: one requiring paracentesis and medical intervention with a D_h_ of 139 Gy (lobar treatment) and one requiring only medical intervention with a D_h_ of 128 Gy (whole liver treatment). Eighteen patients (25%) exceeded the presumed D_h_ 75 Gy threshold, of which eight patients (11%) developed a hepatotoxicity score of ≥2. The remaining 6/14 patients with a hepatotoxicity score of ≥2 had a median D_h_ of 54 Gy (range 19–73 Gy).

In the NELM group, no increased hormone-related complaints (e.g., carcinoid crisis) or infectious problems (e.g., liver abscess or cholangitis) were encountered. Four patients (6%), all without prior biliary interventions, developed a liver abscess at the site of a treated tumor, requiring intravenous antibiotic treatment and additional drainage or right-sided hepatectomy.

In six patients (9%), complaints probably related to the prophylaxis were reported (i.e., diarrhea following ursodeoxycolic acid or diabetic dysregulation during prednisolone use), resulting in early termination of P/UDCA.

#### 3.1.4. Efficacy of P/UDCA

Whole liver treatment and D_h_ showed significant collinearity (*p* < 0.001), as well as previous treatments and tumor type (e.g., chemotherapy and CRC versus NELM and PRRT). To this end, whole liver treatment, previous chemotherapy, and previous PRRT were excluded as variables in subsequent logistic regressions. The remaining non-correlated variables were as follows: WHO performance score (0 or ≥1); tumor type (CRC or NELM); liver tumor burden (median-based: ≤15% vs. >15%); D_h_ (median-based: ≤58 Gy vs. >58 Gy); and P/UDCA (yes or no).

Table 4 shows the results of all analyses. In the analyses, P/UDCA did not show any statistically significant relationship with any of the outcome measures (i.e., summed CTCAE clinical toxicity, summed CTCAE biochemical toxicity, or hepatotoxicity score). None of the investigated variables showed a significant relationship with the summed CTCAE clinical toxicity (Table 4). Only D_h_ showed a significant relationship with summed CTCAE biochemical toxicities in both uni- and multivariate regression (*p* = 0.05). Liver tumor burden showed a significant relationship with the summed biochemical toxicities in only the multivariate analysis (*p* = 0.04). 

### 3.2. Literature Search

The literature search yielded eight relevant studies pertaining to a total of 575 radioembolizations in 534 patients treated with either ^90^Y-glass or ^90^Y-resin microspheres (Figure 1) [14,18,19,20,21,22,23,24]. 

Four studies used medication prophylactically and therapeutically; four studies used medication only therapeutically in response to adverse events. The most common PPMs were proton pump inhibitors, analgesic medications, and steroids (Table 5). Clinical adverse events grade 3 or higher occurred in 63 of 534 patients (11.8%).

There was no similarity concerning PPM between the studies. Each treatment center differed in medication types, the duration of administration, and indications for the PPM. The quality of the included articles was variable. Three studies mentioned the indication for PPM: for the prevention of a carcinoid crisis, hepatobiliary infections, or REILD [14,18,24]. Two studies mentioned the actual names of the administered medications and specifically attempted to investigate the efficacy of the administered PPM [14,24]. One study specifically mentions that patients had no concurrent chemotherapy [22]. Three studies had no standardized way of reporting adverse events (i.e., CTCAE) [18,19,21]. Only one of the included articles had a clear comparative cohort [14], and none of the included articles reported adverse events related to PPM. 

Three publications gave advice concerning PPM. First, Lim et al., although not having used PPM in their study, advised adhering to the manufacturer’s recommended prophylactic H2-antagonist administration [21]. Paradoxically, the study mentioned that no risk factors for toxicity were found in their analysis. Second, Devulapalli et al. retrospectively researched the risk of hepatobiliary infections in patients with a hepatobiliary history being treated with radioembolization and prophylactic use of antibiotics and bowel preparation. A wide variation of prophylactic antibiotic treatment combinations was described and also contained a group not receiving any prophylaxis. The authors concluded that bowel preparation and antibiotic prophylaxis were not associated with a lower risk of infection [24]. Thirdly, Gil-Alzugaray et al. described a treatment protocol to reduce the number of REILD events (defined as bilirubin ≥3 mg/dL (≥51.3 µmol/L) and the presence of ascites clinically or on imaging). Their ‘modified protocol’ incorporated the use of prophylactic medication (ursodeoxycolic acid twice daily 300 mg and methyl prednisolone 8 mg daily for one month followed by 4 mg daily for the subsequent month) and a 10–20% reduction in the calculated activity in whole liver treatments (up to 0.8 GBq/L in cirrhotic patients), while for selective treatments the partition model was used (with a target dose to the non-tumorous liver of 40 Gy in poor candidates, i.e., patients with cirrhosis or extensive pre-treatment with chemotherapy). The authors suggested that with this PMM protocol, the number of REILD cases can be reduced compared to their non-matched historical cohort [14]. However, with the significant reduction in administered activity (in both cirrhotic and non-cirrhotic patients), the contribution of PPM to the decline in REILD is unclear. In their multivariate analysis, both methyl-prednisolone and UDCA use was not associated with REILD. 

Each treatment center in these publications had different protocols concerning patient hospitalization after treatment and adverse event recording. There was no mention of specific patient characteristics that influenced these protocols, apart from the study by Gil-Alzugaray et al., who were more cautious with patients with underlying cirrhosis or previous chemotherapy. 

None of the studies evaluated the post-procedural dosimetric data.

## 4. Discussion

As a start to generating supporting data for PPM, this retrospective cohort study was conducted to investigate the effect of prednisolone and ursodeoxycholic acid (P/UDCA) after radioembolization to prevent REILD (i.e., hepatotoxicity score [17]), in line with the intention of and adopted from Gil-Alzugaray et al. [14]. This study showed that in patients with CRLM or NELM treated with radioembolization, the use of P/UDCA as PPM or REILD-prophylaxis is not supported by conclusive evidence. The only probable variable that correlated with observed toxicity was the whole non-tumorous liver absorbed dose (D_h_). A review of the available literature revealed a wide variety of PPM uses; however, no study provided firm scientific evidence supporting specific PPM uses. No guidelines mention the use of PPM. In line with the findings of previously published international questionnaires, no standardized PPM protocol exists for radioembolization [11,13]. 

Gil-Alzugaray et al. treated patients with ^90^Y resin microspheres, investigating the effect of a so-called ‘modified protocol’: a combination of dose reduction plus prophylaxis with P/UDCA as PPM. Their modified protocol was compared to a non-matched historical group (without prophylaxis or dose reduction). In the multivariate analysis, in non-cirrhotic patients, the modified protocol showed a statistically significant reduction in REILD cases, while methylprednisolone and ursodeoxycholic acid separately were no significant factors. No dosimetric data were available. Yet, the activity/target volume for whole liver treatments was significantly lower in the modified protocol group (1 vs. 0.77 GBq/L; *p* = 0.003) [14]. Interestingly, in their multivariate analysis, the modified protocol did not reduce the number of REILD cases in cirrhotic patients. Furthermore, nearly all REILD cases developed in patients with an activity/target volume of >0.8 GBq/L, which insinuates a significant effect of D_h_.

The current cohort study and the study by Gil-Alzugaray et al. (whole liver treatments in non-cirrhotic patients in 57.8% and 63.4%, respectively) both failed to conclusively prove the usefulness of P/UDCA as PPM following radioembolization to prevent REILD. Both studies, however, do insinuate that D_h_ is a more important variable.

In this study, patients with an HCC (and potential underlying cirrhosis) were intentionally excluded from the study population to prevent confounding of the dosimetric and toxicity data. Though radioembolization is an important treatment strategy for HCC, the value of P/UDCA as prophylaxis in patients with underlying cirrhosis is still to be determined. 

Seidensticker et al. reported that the use of pentoxifylline, UDCA, and low molecular weight heparin as prophylaxis after interstitial brachytherapy reduced the extent of radiation-induced liver damage on hepatobiliary phase imaging at six weeks post-treatment (after 12 weeks, this difference was no longer observed) [25]. After radioembolization, using the same prophylaxis, no significant difference in hepatotoxicity was reported [26]. Their definition of hepatotoxicity was non-standard and highly variable (based on bilirubin and/or ascites or imaging characteristics), and no control group without prophylaxis was selected in the radioembolization study. The value of this prophylactic extended regimen remains unclear. 

Another interesting observation in our cohort was the development of liver abscesses following radioembolization in four patients, as this only occurred in patients without any prior biliary intervention. Contra-intuitive to daily practice, liver abscess formation is feared in patients with a biliodigestive anastomosis, based on high occurrence rates after TACE (even under antibiotic prophylaxis) [23]. This finding was indirectly supported by others, who did provide evidence for an increased risk of infectious complications in patients with prior biliary intervention but without supporting evidence for the use of antibiotic prophylaxis or bowel preparation [27]. Based on these studies and the sole occurrence of liver abscesses in non-biliary-compromised patients in this cohort, antibiotic prophylaxis in patients with prior biliary intervention remains unsupported.

In this study, the summed CTCAE grades of adverse events were used instead of individual CTCAE grades. As Table 3 shows, most patients either experienced no or only grade 1 clinical adverse events/biochemical toxicities. There were some cases of grade ≥2 toxicity, but not enough to form a reliable sample size for accurate analysis per adverse event/biochemical toxicity. Hepatotoxicity, as described by Braat et al., was also used to give a more holistic representation of the effect of radioembolization on the patient and REILD [17]. It also provides a clear definition of REILD using clinical and biochemical parameters, as histopathological correlation is lacking and biochemical toxicity on its own does not reflect the actual loss of liver function. The number of serious clinical, biochemical, and hepatotoxic adverse events in our population was limited, which hampered the analysis of PPM use on the one hand, but further questioned the use of PPM to begin with on the other hand. 

The relatively small number of studies found on this topic in the literature convey that the efficacy of PPM in patients being treated with radioembolization is not a broadly researched subject and that there is no consistent trend in PPM protocols across treatment centers. The results of this retrospective cohort study and the literature search seem to be in line with the advice from Lim et al. and Devulapalli et al. that PPM has not proven to be effective in protecting against adverse events due to radioembolization.

The retrospective cohort study had several limitations, besides being a retrospective analysis, limiting the power to establish a reliable causal relationship, as there may have been unknown confounders left out of the regression model. Furthermore, the relatively small sample size (*n* = 70) limited the statistical power of our analyses. However, the structured practice did allow for a limited amount of missing data and consistent timing of follow-up, which has been unchanged for years. Also, the carefully selected patient population (only two tumor types and one microsphere type (^90^Y glass)) allowed for consistent and comparable post-procedural dosimetric analysis. 

A limitation of the literature search is the limited number of articles discussing PPM and the lack of recent data (most studies were conducted before 2010). They should be considered outdated, as the field of radioembolization has rapidly evolved over the past 10 years.

Future (prospective) studies on the toxicity of radioembolization should at least take note of PPM and important dosimetric values (i.e., D_h_). Preferably, more data are gathered to investigate the efficacy of PPM in general, also considering the side-effects of PPM. As proper scientific evidence supporting the use of steroids, ursodeoxycholic acid, proton-pump inhibitors, octreotide infusion in neuroendocrine tumor patients, and antibiotics in patients with a history of biliary interventions is lacking, the use of any PPM cannot be advised or supported at the moment in patients with metastatic disease.

## 5. Conclusions

No standardized international guidelines or proper supporting evidence exist for any periprocedural medication in radioembolization. The use of prednisolone and ursodeoxycholic acid as prophylaxis was not supported. The whole non-tumorous liver-absorbed dose was the only significant factor for hepatotoxicity.

## Figures and Tables

**Figure 1 diagnostics-13-03652-f001:**
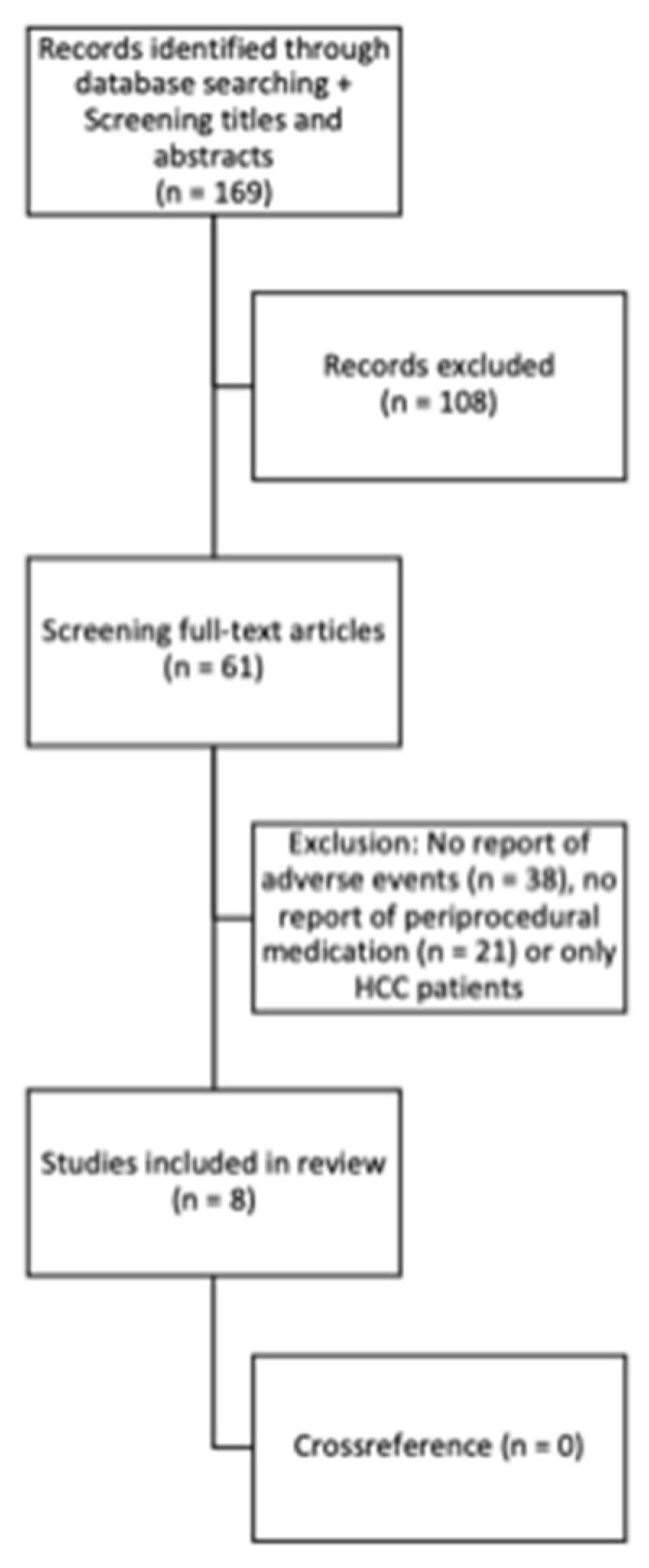
Literature search.

**Table 1 diagnostics-13-03652-t001:** Baseline characteristics of 70 patients.

Characteristics	
**Median Age (range)**	64 (42–86)
Female	24 (34%)
Male	46 (66%)
**Tumor type**	
CRLM	40 (57%)
NELM	30 (43%)
NET grade 1	10 (33%)
NET grade 2	11 (37%)
NET grade 3	5 (17%)
NET grade unknown	4 (13%)
**WHO performance score**	
0	42 (60%)
1	25 (36%)
2	2 (3%)
3	1 (1%)
**Diabetes mellitus**	10 (14%)
**Previous therapies**	
^177^Lu-DOTATATE PRRT *	19 (27%)
4 cycles	7 (10%)
6 cycles	9 (13%)
8 cycles	2 (3%)
**Chemotherapy**	
1 line	14 (20%)
2 lines	15 (21%)
≥3 lines	13 (19%)
Embolotherapy	6 (9%)
Ablation ^†^	17 (24%)
Right sided hepatectomy	2 (3%)
Left sided hepatectomy	2 (3%)
Metastasectomy ^‡^	14 (20%)
SBRT	2 (3%)
**Radioembolization approach**	
Whole liver	38 (54%)
Lobar	24 (34%)
Segmental	8 (11%)
Interval calibration—therapy	
≤7 days	37 (53%)
>7 days	33 (47%)
**Liver volume (mL)**	
Median total liver (range)	1841 (821–4261)
Median healthy parenchyma (range)	1554 (787–3284)
Median tumor (range)	230 (0.4–1980)
Median tumor involvement in % (range)	15 (0.1–48)
**Dosimetry**	
Median prescribed average absorbed dose in Gy (range)	120 (30–300)
Median total ^90^Y Activity in GBq (range)	2.7 (0.3–9.5)
Median D_h_ in Gy (range)	58 (5–139)
D_h_ > 75 Gy	18 (26%)

CRLM = colorectal liver metastasis; NELM = neuro-endocrine liver metastasis; NET = neuroendocrine tumor; PRRT = peptide receptor radionuclide therapy; SBRT = stereotactic body radiotherapy; WHO = world health organization; MIRD = medical internal radiation dosimetry; D_h_ = absorbed dose in the total non-tumorous liver. * In NELM patients only. ^†^ In total, 45 ablations (range 1–6) in 17 patients, either radiofrequency or microwave ablation. ^‡^ In total, 26 metastasectomies (range 1–6) in 14 patients.

**Table 2 diagnostics-13-03652-t002:** Prophylactic periprocedural medication combinations in a retrospective cohort (*n* = 70).

Pre-Procedural Medication	*n*	%
Dexamethasone + Ondansetron	61	87%
Dexamethasone only	1 *	1%
Ondansetron only	3 ^†^	4%
None	5	7%
**Postprocedural medication**		
Pantoprazole	62	89%
Prednisolone + ursodeoxycholic acid	51	73%
Ursodeoxycholic acid only	3	4%
None	8	11%
**All pre- and postprocedural medication**	51	73%
**No medication at all**	5	7%

* Refrained from ondansetron because of pre-existing ECG abnormalities. ^†^ Refrained from dexamethasone because of diabetes mellitus. The following dosages were used: dexamethasone 8 mg once, two hours before intervention; ondansetron 8 mg once, two hours before intervention; prednisone 10 mg daily for one month; subsequently, prednisone 5 mg daily for one month; ursodeoxycholic acid 300 mg twice a day for two months; and pantoprazole 40 mg once daily for six weeks.

**Table 3 diagnostics-13-03652-t003:** Adverse events according to CTCAE grading.

CTCAE Grade	1	2	3	4	5
**Clinical toxicity**					
Fatigue	55%	16%			
Abdominal pain	38%	7%	2%		
Other pain	4%	3%			
Nausea	37%	4%			
Vomiting	7%	3%			
Malaise	10%	2%			
Fever	10%	2%			
Loss of appetite	17%	4%			
**Biochemical toxicity ***					
Bilirubin	4%	2%	6%		
Alkaline phosphatase	44%	6%	3%		
Gamma-glutamyltransferase	39%	28%	11%	2%	2%
Aspartate aminotransferase	25%	6%			
Alanine aminotransferase	11%	4%			
Albumin	14%	3%	2%		
Lactate dehydrogenase	45%				
**Complications**					
Abscess				6%	
Clinical progressive disease			4%	6% ^†^	
REILD ^‡^				3%	
Radiation cholecystitis			2%		
**Hepatotoxicity**	24%	17%	1%	1%	

REILD = radioembolization-induced liver disease. * Complete laboratory tests at 3 months follow-up missing for four patients (6%), ^†^ Requiring paracentesis because of peritonitis carcinomatosis-induced ascites. Progressive disease confirmed on imaging studies in all patients. ^‡^ Overlap with hepatotoxicity score [13].

**Table 4 diagnostics-13-03652-t004:** Logistic regression models.

Clinical Toxicity	Univariate				Multivariate			
** *Variables* **	** *B* **	** *p* **	** *OR* **	** *95% CI* **	** *B* **	** *p* **	** *OR* **	** *95% CI* **
WHO	0.20	0.73	1.22	0.4–3.8	0.56	0.40	1.74	0.5–6.3
Tumor type	−1	0.08	0.35	0.1–1.1	−0.68	0.29	0.51	0.1–1.8
Liver burden	1.09	0.07	2.97	0.9–9.7	0.82	0.21	2.27	0.6–8.1
D_h_	−0.33	0.57	0.72	0.2–2.2	−0.42	0.50	0.66	0.2–2.2
LDT	−1.09	0.12	0.34	0.1–1.3	−1.15	0.14	0.32	0.1–1.4
P/UDCA	−0.15	0.82	0.86	0.2–3.1	−0.21	0.77	0.81	0.2–3.6
**Biochemical toxicity**								
** *Variables* **	** *B* **	** *p* **	** *OR* **	** *95% CI* **	** *B* **	** *p* **	** *OR* **	** *95% CI* **
WHO	−0.71	0.27	0.49	0.1–1.7	−0.84	0.25	0.43	0.1–1.8
Tumor type	0.11	0.86	1.11	0.3–3.6	−0.21	0.76	0.81	0.2–3.2
Liver burden	−1.16	0.07	0.31	0.1–1.1	**−1.45**	**0.04 ***	**0.24**	**0.1–1.0**
D_h_ ^†^	**1.31**	**0.04 ***	**3.71**	**1.0–13.1**	**1.37**	**0.05 ***	**3.92**	**1.0–15.0**
LDT	−0.16	0.79	0.85	0.3–2.8	0.23	0.75	1.26	0.3–5.3
P/UDCA	0.13	0.85	1.14	0.3–4.7	0.04	0.96	1.04	0.2–5.0
**Hepatotoxicity**								
** *Variables* **	** *B* **	** *p* **	** *OR* **	** *95% CI* **	** *B* **	** *p* **	** *OR* **	** *95% CI* **
WHO	−1.08	0.12	0.34	0.1–1.3	−0.80	0.30	0.45	0.1–1.9
Tumor type	−0.72	0.23	0.49	0.1–1.6	−0.43	0.52	0.65	0.2–2.4
Liver burden	−0.43	0.48	1.54	0.5–5.0	0.22	0.74	1.25	0.4–4.6
D_h_	0.73	0.24	2.10	0.6–7.0	0.46	0.49	1.59	0.4–5.9
LDT	**−2.28**	**0.03 ***	**0.10**	**0.0–0.8**	−1.97	0.07	0.34	0.1–1.2
P/UDCA	0.69	0.40	2.00	0.4–10.1	0.84	0.34	2.32	0.4–13.1

WHO = World Health Organization performance score; D_h_ = whole healthy liver absorbed dose; LDT = liver directed treatment (i.e., whole liver: yes/no). * Indicates significance, *p* < 0.05. ^†^ D_h_ is the only significant factor in both uni- and multivariate logistic regression for biochemical toxicity, *p* < 0.05. In subsequent non-dichotomized (continued variables) univariate and multivariate logistic regressions, only D_h_ remains a significant factor: per 1 Gy D_h_, the likelihood of developing >4 summed CTCAE biochemical toxicities increases by 3% (*p* = 0.018).

**Table 5 diagnostics-13-03652-t005:** Literature search results.

Author (Year)	Product	*n* (*p*)	Tumor Types	Cirrhosis	Activity Calculation Method	GBq (Range)	PPM	Comments and Author’s Advice
King et al. (2008) [14]	Resin	34 (46)	NELM	NR	BSA	Mean 1.99 (0.92–2.80)	Prophylactic: Octreotide + H2-antagonist for 1 month	No advice regarding PPM. No patients were coiled.
Stubbs et al. (2001) [15]	Resin	50 (50)	CRLM	NR	NR	Mean 2.27 (2.0–3.0)	Narcotics	No advice regarding PPM.
Murthy et al. (2005) [16]	Resin	12 (17)	CRLM	NR	BSA and Empirical	Mean 1.47 (0.63–2.5)	Antiemetics, narcotic analgesics, supportive care, parenteral antibiotics	No mention of prophylactic medication or efficacy of periprocedural medication.
Lim et al. (2005) [17]	Resin	29 (29)	CRLM	NR **	BSA	NR	Antiemetics, analgesics	Prophylactic H2-antagonist recommended. No risk factors for toxicity were found.
Pöpperl et al. (2005) [18]	Resin	23 (23)	Mixed (7/23 CRLM/1/23 NELM)	NR	Empirical	Mean 2.5 (NR)	Analgesics, antiemetics, steroids, drugs for gastric protection and AB	No mention of the efficacy of PPM or advice concerning usage of PPM.
Gil-Alzugaray et al. (2013) [10]	Resin	260 (260)	Mixed (67/172 CRLM/25/172 NELM)	34% (172 non-cirrhotic)	BSA and partition modelling	Mean NR (0.6–2.23)	Prophylactic: Ursodeoxycolic acid and Methyl prednisolone	Prescribed activity was reduced for all subgroups with ‘modified protocol’ (partition modelling and prophylaxis), and reduced the occurrence of REILD. In MVA, occurrence of REILD was reduced by the ‘modified protocol’.
Cholapranee et al. (2015) [19]	Resin	SIRT 16 (24) * TACE 13 (24)	Mixed (1/16 CRLM/9/16 NELM)	NR	BSA	NR	Extensive AB prophylaxis: oral levofloxacin 500 mg daily + metronidazole 500 mg twice daily	No control group without AB prophylaxis. With AB no infectious complication in the SIRT group. 23% liver abscesses in the TACE group.
Devulapalli et al. (2018) [20]	Glass and resin	126 (92 glass, 88 resin) ^†^	Mixed (69% metastatic disease)	NR	MIRD and BSA	Glass: Median 2.36 (NR); Resin: Median 1.04 (NR)	Prophylactic: AB in 79%, various types; most common: levofloxacine + metronidazole (43%)	Incidence of liver abscess is rare (7.9%). Bowel preparation and antibiotic prophylaxis were not associated with lower risk of infection.

*n* = number of patients; *p* = number of treatments; BSA = body surface area; NELM = neuroendocrine liver metastases; CRLM = colorectal liver metastases; AB = antibiotic prophylaxis; SIRT = selective internal radiation therapy, a.k.a. radioembolization; NR = Not reported. * Including 11 patients with a biliodigestive anastomosis (either pancreatic adenocarcinoma or cholangiocarcinoma as the primary tumor) and 5 patients with a biliary stent placement. ^†^ Including 54 repeated treatments (in 47 patients), all with the same microsphere type as the initial treatment. ** Patients with liver decompensation or portal hypertension were excluded.

## Data Availability

Data is contained within the article and Appendix A.

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
