# Peer review of "Prophylactic Medication during Radioembolization in Metastatic Liver Disease: Is It Really Necessary? A Retrospective Cohort Study and Systematic Review of the Literature"

_diagnostics, 2023, doi:10.3390/diagnostics13243652_

Round 1

Reviewer 1 Report

Comments and Suggestions for Authors

The work under evaluation examines the scientific evidence regarding the effectiveness of pre- and peri-procedural prophylaxis in preventing toxicity related to TARE, particularly with respect to REILD. The work consists of two parts: 1) a retrospective study with two cohorts; 2) a systematic literature review. Regarding the retrospective study, patients with liver metastases from colorectal or neuroendocrine tumors undergoing treatment with 90Y-glass microspheres were included. Various factors, including prophylaxis and the dose received by healthy parenchyma, were assessed as predictors of toxicity. Fifty-one patients received P/UDCA 20 as post-treatment medication, while 19 did not. No correlation was found between toxicity and P/UDCA use. Conversely, the dose to the healthy liver (DH) was found to be correlated with biochemical toxicity. In light of the cohort study data, supported by the systematic review, there is no evidence to support the use of prophylaxis in patients treated with TARE.

The study is interesting, well-conducted, and provides clinically relevant information. Some considerations:

- The authors should emphasize in the Discussion that their cohort did not include patients with HCC-related cirrhosis, in whom potential prophylaxis should be separately assessed due to the increased risk of liver failure.

- Some parameters, such as NLR or PLR, have emerged as important prognostic stratifiers in patients with neoplasms and also in subjects treated with TARE. Specifically, an increased NLR value indicates an inflammatory and pro-coagulatory microenvironment. In these cases, might it be useful to investigate the utility of prophylaxis, especially with corticosteroids and pentoxifylline? Please elaborate, adding a brief sentence in the Discussion, citing the following studies DOI:10.1245/s10434-014-4050-6, DOI:10.1097/MNM.0000000000001123, DOI: 10.3978/j.issn.2078-6891.2015.108

Author Response

Dear reviewer 1,

Thank you for your thorough review of our manuscript: ‘Prophylactic Medication During Radioembolization in Metastatic Liver Disease, it it Really Necessary? A retrospective Cohort Study and Review of Literature’. We took the time to revise this manuscript according to your comments. A detailed point-by-point response to your comments is given below, and the changes in the revised manuscript are highlighted in yellow.

Hopefully, we have answered your questions satisfactory.

The study is interesting, well-conducted, and provides clinically relevant information. Some considerations:

- The authors should emphasize in the Discussion that their cohort did not include patients with HCC-related cirrhosis, in whom potential prophylaxis should be separately assessed due to the increased risk of liver failure.

We agree that this deserves some extra attention. Therefore, we added the following brief paragraph to the discussion (highlighted in the document):

In this study patients with an HCC (and potential underlying cirrhosis) were intentionally excluded from the study population to prevent confounding of the dosimetric and toxicity data. Though radioembolization is an important treatment strategy for HCC, the value of P/UDCA as prophylaxis in patients with underlying cirrhosis is still to be determined.

- Some parameters, such as NLR or PLR, have emerged as important prognostic stratifiers in patients with neoplasms and also in subjects treated with TARE. Specifically, an increased NLR value indicates an inflammatory and pro-coagulatory microenvironment. In these cases, might it be useful to investigate the utility of prophylaxis, especially with corticosteroids and pentoxifylline? Please elaborate, adding a brief sentence in the Discussion, citing the following studies DOI:10.1245/s10434-014-4050-6, DOI:10.1097/MNM.0000000000001123, DOI: 10.3978/j.issn.2078-6891.2015.10

Thank you for the suggestion. We reviewed the references you have provided us with. Most studies on NLR and PLR report the prognostic value of the baseline NLR and/or PLR for overall survival or progression-free survival. Only one of the studies reported an analysis with regard to the adverse events. Tohme et al. reported that common procedure-related adverse events (fatigue, nausea, and/or vomiting, abdominal pain, fever, and increased bilirubin) were predominately mild-to-moderate in intensity and short in duration, and had no statistically significant difference between the groups with a NLR <5 and NLR>5 at baseline were observed (data not shown). The ΔNLR or PLR was not reported.

Furthermore, in our study on the NLR and PLR in NELM patients after 166Ho radioembolization an increase in NLR and PLR at three weeks follow up was significantly associated with response on imaging (at 3, 6, 9 and 12 months) (Ebbers et al. EJNMMI research 2022). Though not reported, no correlation was observed with the hepatotoxicity in this study population. None of the patients in that study were included in our cohort.   

We decided not to include a paragraph or brief sentence on the NLR or PLR, as its value with regard to the hepatotoxicity is unclear. As mentioned in the discussion we believe “the value of this prophylactic extended regimen remains unclear”, also with regard to the NLR and PLR.

Best regards,
Manon Braat, on behalf of all authors

Reviewer 2 Report

Comments and Suggestions for Authors

1,Please check if the expression is clear or in line with habit

Retrospective analyses included patients with progressive liver dominant or liver-only colorectal liver metastases (CRLM) or neuroendocrine liver metastases (NELM)......liver dominant and CRLM or NELM?

From 2016 to mid-2019, based on a previous publication from Gil-Alzugaray et al., P/UDCA was standard care in our center, consisting of ursodeoxycholic acid 600 mg daily for two months and prednisone 10 mg daily for one month, followed by 5 mg daily for the second month; all starting the day of treatment: followed by 5 mg daily for the second month, I think it may be about the dosage of prednisone?

2, 3.1.4. Efficiency of P/UDCA section: Should the author use a table to describe the statistical results.

 3,The conclusion section: Whole non tumorous liver absorbed do was the only significant factor for pathotoxicity. Move to the front of the section. The Abstract section does not have this conclusion. Would you like to add it.

Comments on the Quality of English Language

Overall, this is a good written and conducted manuscript with a topic of current interest. I have a few of comments the authors should be aware of. 

1,Please check if the expression is clear or in line with habit

Retrospective analyses included patients with progressive liver dominant or liver-only colorectal liver metastases (CRLM) or neuroendocrine liver metastases (NELM)......liver dominant and CRLM or NELM?

From 2016 to mid-2019, based on a previous publication from Gil-Alzugaray et al., P/UDCA was standard care in our center, consisting of ursodeoxycholic acid 600 mg daily for two months and prednisone 10 mg daily for one month, followed by 5 mg daily for the second month; all starting the day of treatment: followed by 5 mg daily for the second month, I think it may be about the dosage of prednisone?

2, 3.1.4. Efficiency of P/UDCA sectionShould the author use a table to describe the statistical results.

 3,The conclusion section: Whole non tumorous liver absorbed do was the only significant factor for pathotoxicity. Move to the front of the section. The Abstract section does not have this conclusion. Would you like to add it.

Author Response

Dear reviewer 2,

Thank you for your thorough review of our manuscript: ‘Prophylactic Medication During Radioembolization in Metastatic Liver Disease, it it Really Necessary? A retrospective Cohort Study and Review of Literature’. We took the time to revise this manuscript according to your comments. A detailed point-by-point response to your comments is given below, and the changes in the revised manuscript are highlighted in yellow.

Hopefully, we have answered your questions satisfactory.

1,Please check if the expression is clear or in line with habit:

Retrospective analyses included patients with progressive liver dominant or liver-only colorectal liver metastases (CRLM) or neuroendocrine liver metastases (NELM)......liver dominant and CRLM or NELM?

We added “with” to the sentence. Patients had either liver dominant or liver-only colorectal metastases OR neuroendocrine liver metastases. In the NELM group also patients with extrahepatic disease were included. The extrahepatic disease was regarded as optimally treated with PRRT. NELM respond less well to PRRT and often are the prognosis defining metastases.

From 2016 to mid-2019, based on a previous publication from Gil-Alzugaray et al., P/UDCA was standard care in our center, consisting of ursodeoxycholic acid 600 mg daily for two months and prednisone 10 mg daily for one month, followed by 5 mg daily for the second month; all starting the day of treatment: followed by 5 mg daily for the second month, I think it may be about the dosage of prednisone?

This is correct. We changed it to “followed by prednisone 5 mg daily”.
We also changed this in the legend of Table 2.

2, 3.1.4. Efficiency of P/UDCA section: Should the author use a table to describe the statistical results.

We described the statistical results in Supplemental Table 3. We now added this table the manuscript (now Table 4) and changed the numbering of the tables. These changes are highlighted in yellow.

 3,The conclusion section: Whole non tumorous liver absorbed do was the only significant factor for pathotoxicity. Move to the front of the section. The Abstract section does not have this conclusion. Would you like to add it.

We added “Whole non-tumorous liver absorbed dose was the only significant factor for hepatotoxicity to the conclusion of the abstract.

Best regards,
Manon Braat, on behalf of all authors

Reviewer 3 Report

Comments and Suggestions for Authors

I’m glad to have the opportunity to review this paper diagnostics-2711565 ‘Prophylactic Medication During radioembolization in Metastatic Liver Disease, it it Really Necessary? A retrospective Cohort Study and Review of Literature’. Although transarterial radioembolization (TARE) is a well-known treatment for liver malignancies, there are no consensus and evidence regarding periprocedural prophylactic medication (PPM). TARE is generally in the spotlight as a treatment for HCC, but it can also be used to treat intrahepatic metastases of other carcinomas.

This paper provides a good description of PPM in patients who chose TARE for the treatment of intrahepatic metastases of carcinomas other than HCC. Although the target patient group is small, it is considered to be valuable because TARE is difficult to select as a treatment for intrahepatic metastatic cancer. They also provided an additional literature review and organized it well. However, since TARE is generally used to treat HCC, I ask that further description of PPM in HCC will enrich the content.

Therefore, I recommend that this paper is suitable for publication in this journal as long as additional content is provided.

Author Response

Dear reviewer 3,

Thank you for your thorough review of our manuscript: ‘Prophylactic Medication During Radioembolization in Metastatic Liver Disease, it it Really Necessary? A retrospective Cohort Study and Review of Literature’. We took the time to revise this manuscript according to your comments. A detailed point-by-point response to your comments is given below, and the changes in the revised manuscript are highlighted in yellow.

Hopefully, we have answered your questions satisfactory.

I’m glad to have the opportunity to review this paper diagnostics-2711565 ‘Prophylactic Medication During radioembolization in Metastatic Liver Disease, it it Really Necessary? A retrospective Cohort Study and Review of Literature’. Although transarterial radioembolization (TARE) is a well-known treatment for liver malignancies, there are no consensus and evidence regarding periprocedural prophylactic medication (PPM). TARE is generally in the spotlight as a treatment for HCC, but it can also be used to treat intrahepatic metastases of other carcinomas.

This paper provides a good description of PPM in patients who chose TARE for the treatment of intrahepatic metastases of carcinomas other than HCC. Although the target patient group is small, it is considered to be valuable because TARE is difficult to select as a treatment for intrahepatic metastatic cancer. They also provided an additional literature review and organized it well. However, since TARE is generally used to treat HCC, I ask that further description of PPM in HCC will enrich the content.

Therefore, I recommend that this paper is suitable for publication in this journal as long as additional content is provided.

Thank you for your careful assessment and suggestions.
We agree that radioembolization is an important treatment strategy for HCC. However, in the current EANM guidelines the recommended absorbed liver dose for cirrhotic patients is considerably lower (for all commercially used microspheres). Therefore, we excluded these patients from our analysis (i.e. to prevent confounding of the dosimetric and toxicity data).

Our initial literature search yielded one study on prophylaxis in a HCC population (
Salem et al. Radioembolization results in longer time-to-progression and reduced toxicity compared with chemoembolization in patients with hepatocellular carcinoma. Gastroenterology. 2011). This study mentioned the use of a five day PPI treatment following radioembolization, but no data on the use of P/UDCA. After careful consideration, we decided not to include these data in the analysis of the literature and to focus on the non-cirrhotic patient populations, similar to our study population.
Given the lack of evidence for prophylaxis in the cirrhotic (and non-cirrhotic) populations we feel we cannot discuss the value of P/UDCA more in depth for HCC patients or cirrhotic patients.

Therefore, we added the following brief paragraph to the discussion (highlighted in the document):

In this study patients with an HCC (and potential underlying cirrhosis) were intentionally excluded from the study population to prevent confounding of the dosimetric and toxicity data. Though radioembolization is an important treatment strategy for HCC, the value of P/UDCA as prophylaxis in patients with underlying cirrhosis is still to be determined.

Best regards,
Manon Braat, on behalf of all authors